# Outcomes of Corneal Transplantation for Herpetic Keratitis: A Narrative Review

**DOI:** 10.3390/v16091403

**Published:** 2024-08-31

**Authors:** Michele Nardella, Angeli Christy Yu, Massimo Busin, Roberta Rizzo, Giorgio Zauli

**Affiliations:** 1Department of Translational Medicine, University of Ferrara, 44121 Ferrara, Italy; michele.nardella@unife.it (M.N.); angelichristy.yu@unife.it (A.C.Y.); massimo.busin@unife.it (M.B.); 2Department of Ophthalmology, Ospedali Privati Forlì “Villa Igea”, 47122 Forlì, Italy; 3Istituto Internazionale per la Ricerca e Formazione in Oftalmologia (IRFO), 47122 Forlì, Italy; 4Department of Chemical, Pharmaceutical and Agricultural Sciences, 44121 Ferrara, Italy; roberta.rizzo@unife.it; 5Research Department, King Khaled Eye Specialist Hospital, Riyadh 12329, Saudi Arabia

**Keywords:** cornea, herpes simplex virus (HSV), herpetic keratitis, penetrating keratoplasty (PK), deep anterior lamellar keratoplasty (DALK), mushroom keratoplasty (MK)

## Abstract

Herpes simplex virus (HSV) is one of the most common etiologic agents of corneal disease and a significant cause of corneal blindness worldwide. Although most cases can be successfully managed with medical therapy, HSV keratitis associated with visually significant stromal scarring often requires corneal transplantation for visual rehabilitation. While penetrating keratoplasty (PK) represented the traditional keratoplasty technique, the past few decades have seen a shift towards lamellar keratoplasty procedures, including deep anterior lamellar keratoplasty and mushroom keratoplasty. This paper describes the current surgical techniques and perioperative antiviral prophylaxis regimen for herpetic keratitis and reviews their postoperative clinical outcomes.

## 1. Introduction

Herpes simplex virus (HSV) infection is one of the most common worldwide, with humans being the only known natural host. According to the World Health Organization (WHO), over 4 billion individuals aged 0–49 years are infected, representing a prevalence rate of 79.8% [1]. HSV type 1 (HSV-1) and HSV type 2 (HSV-2) are antigenically related; however, HSV-1 predominantly causes orofacial and ocular infections, while HSV-2 is more commonly associated with genital infections. However, both viruses can cause infections in either region and remain in a latent state in the sensory ganglia [2].

Primary HSV infection typically occurs during childhood, following contact with infected lesions or their secretions. The majority of primary HSV infections are asymptomatic or present with nonspecific oropharyngeal symptoms [3,4].

The most common manifestation of primary ocular HSV infection is moderate to severe conjunctivitis or blepharitis. Around 15% of these patients had epithelial keratitis, and 2% had stromal keratitis as their first ocular presentation [5]. Of note, the most common manifestation of recurrent ocular HSV infection is HSV keratitis. According to the Herpetic Eye Disease Study (HEDS) in the U.S., epithelial keratitis accounted for 47% of cases, while HSV stromal keratitis accounted for 16% [6].

HSV keratitis affecting various corneal layers has distinct pathogenic mechanisms. HSV epithelial keratitis arises from the direct infection of corneal epithelial cells, while the pathogenesis of HSV stromal keratitis is primarily immunologic. Thus, treating HSV dendritic epithelial keratitis primarily involves antiviral medications, whereas HSV stromal keratitis typically necessitates a combination of antiviral drugs and topical corticosteroids [7]. HSV type-1 establishes latency in the trigeminal ganglia following primary infection of the cornea. HSV-1 virus remains dormant in neuronal cells, avoiding immune detection. Reactivation of HSV-1 due to various triggers, such as stress or immunosuppression, can lead to recurrent corneal infections [7]. Control of HSV-1 latency and reactivation is largely mediated by the host’s immune system, particularly through the actions of T-cells (particularly CD8+ T cells) and cytokines that suppress viral gene expression and maintain the virus in a quiescent state [8]. Because of the established latent state, medical treatment for HSV does not eradicate the virus; it only decreases the duration of symptoms and reduces the risk of recurrence.

Recurrence of HSV infection increases the patient’s risk of developing corneal scarring, neovascularization, and blindness. Specifically, the leading cause of corneal blindness in developed countries is HSV stromal keratitis [9]. Corneal scarring, neovascularization, thinning, necrosis, and perforation are the main indications for keratoplasty for HSV keratitis [9,10].

Keratoplasty in eyes with herpetic keratitis is traditionally considered to confer a high risk of graft failure [11]. Reactivation of latent infection, which results in the clinical recurrence of herpetic ocular disease, increases the risk of immunologic rejection. Patients with a history of ocular HSV infection have a higher incidence of allograft rejection compared to those without such a history [12]. Therefore, many corneal surgeons avoid performing corneal transplantation in such high-risk eyes. However, in the last decades, advancements in surgical techniques and instrumentation, along with postoperative antiviral prophylaxis, have significantly enhanced the success rates of keratoplasty [13].

The aim of this review is to describe the current surgical techniques and perioperative antiviral prophylaxis regimen for herpetic keratitis, with special emphasis on the postoperative clinical outcomes.

## 2. Material and Methods

This narrative review includes only original articles and case series whose main aim was to report the outcomes regarding penetrating keratoplasty (PK), deep anterior lamellar keratoplasty (DALK), or mushroom keratoplasty (MK) in patients with corneal scarring due to HSV keratitis. Articles regarding postoperative medical management were also included. Some of the included studies compared these surgical techniques, but articles in which intervention was PK, DALK, or MK without any comparison were also considered. Systematic and narrative reviews were excluded.

A literature search was performed on the PubMed library using the following search terms: (keratoplasty) AND ((herpetic keratitis) OR (HSV keratitis) OR (herpes keratitis)) published between January 1993 and present. The articles were screened based on the aforementioned inclusion/exclusion criteria by title and abstract.

## 3. Discussion

Penetrating keratoplasty (PK) involves the full-thickness transplantation of a donor cornea to replace the damaged corneal tissue (Figure 1A) [14]. On the other hand, deep anterior lamellar keratoplasty involves selective replacement of diseased corneal stroma with the anterior donor lamella (Figure 1B). Mushroom keratoplasty is a form of penetrating keratoplasty that involves minimal endothelial replacement with transplantation of a large 9 mm diameter anterior lamella and a smaller 6 mm diameter posterior lamella (Figure 1C).

Over the last half of the 20th century, PK became the standard of care for the surgical treatment of most corneal diseases, including severe corneal scarring due to HSV keratitis [15]. PK is indicated in eyes with impending or frank corneal perforation due to necrotizing or neurotrophic HSV ulcers [7]. Patients with active inflammation, active ulceration, perforation, or extensive neovascularization secondary to HSV keratitis have been shown to have particularly poor success rates following PK [16].

In a study by Lomholt et al., the recurrence rate of HSV keratitis in patients who received PK with no antiviral prophylaxis was 44% within the first 2 years postoperatively [17]. The rejection rate was 29% during the first year and 46% within 2 years [17].

On the other hand, in a retrospective cohort of 454 patients receiving primary PK for herpetic keratitis reported to NHS Blood and Transplant (NHSBT), oral antiviral treatment has been found to reduce the risk of graft failure [18]. Particularly, patients receiving oral antivirals were less than a third as likely to have a failed graft at 5 years postoperatively compared with those on no antiviral medication (relative risk (RR) 0.3, CI: 0.2–0.7, *p* = 0.002) [18].

In a study by Wu et al. on the outcomes of PK in 58 eyes of 63 patients with corneal scarring due to HSV keratitis who received oral acyclovir 200 mg five times daily in the first 3 months, tapered to 400 mg twice daily for 12 to 18 months, the rate of HSV recurrence was 21% while the immune rejection rate was 41%. Moreover, endothelial rejection was remarkably more common than stromal rejection. Furthermore, 22% of patients developed graft failure over a follow-up of 4.7 ± 2.8 years. The causes for graft failure included immune rejection as most common, followed by recurrence of HSV keratitis [19].

Notably, however, the use of oral acyclovir at a dosage of 400 mg five times daily for the first three months, followed by 400 mg twice daily for the next twelve months, has been found to improve the outcomes of PK. In a comparative study by Altay et al. on the outcomes of PK for HSV keratitis with Group 1 including patients with a quiescent herpetic corneal scar (*n* = 42) and Group 2 including patients who developed a corneal descemetocele or perforation due to persistent epithelial defects without active stromal inflammation (*n* = 13), visual outcomes showed no statistically significant differences between the groups. Moreover, HSV keratitis recurrence was observed in 29% of patients in the quiescent scar group, typically presenting as epithelial or combined epithelial and stromal keratitis, and in 31% of patients in group 2. Graft rejection occurred in 10% of patients in Group 1 and in 23% of those in Group 2 [20].

However, despite advances in PK techniques and the use of topical steroids and antiviral prophylaxis, the problem of allograft rejection remains the leading cause of graft failure following PK for herpetic keratitis. Specifically, immune rejection and endothelial graft failure (often resulting from significant endothelial cell loss during a rejection episode) continue to be the primary reasons for graft failure, as demonstrated in numerous studies [21,22].

Graft failure in PK for herpetic keratitis is most commonly attributed to allograft rejection rather than viral recurrence [12]. However, in the absence of pathognomonic signs such as a dendritic or geographic ulcer, it can be clinically challenging to differentiate between rejection and viral recurrence, which may sometimes coexist. Consequently, interpreting recurrence and rejection rates reported in studies is often difficult [23]. Certainly, immune corneal endothelial rejection is a major problem with high-risk corneal transplant recipients, as HSV keratitis eyes usually involve significant corneal stromal vascularization and inflammation [15].

Table 1 outlines the differential outcomes of studies on PK for HSV keratitis.

### 3.1. Deep Anterior Lamellar Keratoplasty (DALK)

In the last few decades, the introduction of lamellar keratoplasty has changed surgeons’ preferences [24]. Deep anterior lamellar keratoplasty (DALK) is a type of lamellar corneal transplantation that involves selective replacement of the diseased host stroma. DALK is currently preferred for the treatment of corneal stromal disease [25]. In high-risk eyes with herpetic scars, prior inflammation, and extensive corneal vascularization but otherwise healthy endothelium, DALK is considered an optimal surgical approach that reduces the risk of infection recurrence, immune rejection, and graft failure [26].

Currently, the big-bubble technique by Anwar and Teichmann is the most widely used approach for DALK [27]. However, even experienced surgeons may not always be able to succeed using this technique, especially in severe post-herpetic corneal opacities with residual scarring down to the Descemet membrane. In these cases, other direct dissection DALK techniques such as manual dissection, which could leave some residual cornea, can be considered, although final vision may not be as good as after PK [28,29].

Another important complication of stromal herpetic keratitis is corneal thinning [10]. Reduction in stromal thickness leads to corneal irregularity and consequent visual impairment. In these cases, performing DALK, especially via pneumatic dissection, is more challenging. In corneas with significant asymmetry in peripheral thickness and high irregularity, peripheral intrastromal hydration with balanced salt solution in areas of relative thinning can be used to facilitate safe, deep trephination [30].

As previously stated, the main cause of low graft survival rate in PK for HSV corneal scars is allograft rejection [12]. The greater advantage of DALK over PK is that the host corneal endothelium is not subject to immune rejection [24,31]. Lomholt et al. demonstrated that primary allograft rejection following PK for HSV keratitis occurred in 29% of grafts within the first year and 46% within the first two years [17]. On the other hand, studies have shown that stromal rejection is a rare complication of DALK surgery [32,33,34]. Furthermore, recurrence of herpetic keratitis was not observed in any of the patients who were all under antiviral prophylactic therapy with acyclovir (800 mg) three times a day for the first month, twice daily during the second month, and once a day as long-term therapy. Similar studies reported no cases of graft failure [35,36]. In a series of 89 eyes who underwent DALK for herpetic keratitis, Ren et al. reported a corneal stromal graft rejection rate of 4.5% and a HSV keratitis recurrence in 9% of patients despite oral antiviral prophylaxis with acyclovir 400 mg five times daily for 1 month, tapered to 400 mg twice daily for another 12 months [29].

A smaller series including 18 eyes by Awan et al. demonstrated a graft rejection rate of 33% at three years, which still compares favorably to the reported rates following PK [37]. In the same series, the recurrence of HSV keratitis was 5% in the first year, which is significantly less in comparison to those reported for PK [37]. In addition, the 3-year graft survival rate was 83%, which is higher than the 2-year survival rate of 67% as reported by Lomholt et al. [17]. Slightly worse results were obtained by Lyall et al. with a graft survival rate of 72% and a higher HSV recurrence rate (33%), probably due to the use of a lower dose of oral acyclovir prophylaxis (400 mg twice daily) [38]. These observations suggest oral acyclovir may minimize the risk but does not completely eliminate the recurrence of HSV keratitis after DALK. Furthermore, a low dose of acyclovir after surgery with the non-use of topical steroids and antibiotics has also been associated with graft failure or graft melting [36].

A study by Wang et al. evaluated the outcomes of DALK in patients with herpetic stromal keratitis [39]. The recurrence rate of HSV keratitis after DALK was higher in eyes with active disease compared to those in the quiescent phase (18.2% vs. 9.5%). This finding aligns with the theory that corneal grafts have better survival rates in the presence of inactive corneal scarring compared to active keratitis [12].

In a long-term comparison of outcomes of DALK and PK for HSV keratitis, the recurrence rate was 10% in the DALK group, significantly lower than the 20% recurrence rate in the PK group. Additionally, there were no cases of rejection in the DALK group, whereas 41% of patients in the PK group experienced rejection. Furthermore, because topical corticosteroids can usually be discontinued earlier with DALK [15], the incidence of steroid-induced high IOP, glaucoma, and cataract was significantly lower in the DALK group compared to the PK group [22].

The major long-term advantage of DALK over PK relates to long-term preservation of host corneal endothelial cells [31]. Several studies have reported excellent postoperative endothelial cell density even in eyes undergoing DALK for herpetic keratitis in a long follow-up [38,39]. Because of postoperative stability of endothelial cell density, long-term graft survival in DALK is remarkably better than that in the PK. In addition, DALK offers a better safety profile as an extraocular procedure, minimizing the risks associated with open-sky surgery [22].

Regarding visual outcomes, several studies have shown that both PK and DALK lead to similar improvements in visual acuity. However, directly comparing the visual acuity improvement between the two surgical groups is challenging due to the variability in how the results are presented [14]. Li, J. et al. found that 74.1% of eyes undergoing DALK, using either the big bubble technique or manual lamellar dissection and receiving full-thickness stroma, achieved a postoperative best spectacle-corrected visual acuity (BSCVA) of 20/40 or better at a 3-year follow-up visit [36]. Similarly, 71.4% of eyes undergoing DALK with a precut anterior lamellar graft reached this level of visual acuity. Visual outcomes were similar to other DALK reports using the big bubble technique for HSV-related corneal scars [33].

A retrospective interventional case series compared 124 cases of large 9.0-mm DALK with 133 cases of conventional 8.0-mm DALK. The 9.0-mm DALK group had significantly better visual outcomes, with 44% achieving a Snellen BSCVA of 20/20 or better compared to 26% in the 8.0-mm DALK group, and 74% achieving 20/25 or better compared to 59%. Additionally, the 9.0-mm DALK group experienced significantly less astigmatism, with an average postoperative refractive astigmatism that was 1 D lower than those observed in the 8.0-mm group. High-degree astigmatism (>4.5 D) was observed in only 7% of the 9.0-mm DALK cases [40].

In a study comparing the outcomes comparing DALK and converted MK for herpetic corneal scars, DALK was successfully performed in 98 of 120 eyes, while the remaining 22 eyes required intraoperative conversion to MK. At 5 years, mean logMAR BSCVA was 0.10 ± 0.12 in the DALK group and 0.09 ± 0.15 in the MK group (*p* = 0.75). Refractive astigmatism at 5 years was 2.8 ± 1.4 D in the DALK group and 3.0 ± 1.7 D in the MK group (*p* = 0.67). ECD was higher in the DALK group than in the MK group at all time points (*p* < 0.001), with a mean annual cell loss of 10.9% after MK and 4.2% after DALK. The 5-year risk for immunologic rejection (DALK: 3%, MK: 5%, *p* = 0.38), herpetic recurrence (DALK: 6%, MK: 9%, *p* = 0.38), and graft failure (DALK: 4%, MK: 5%, *p* = 0.75) were comparable in both groups [13].

Despite the advantages over PK, DALK is not without its complications. As mentioned above, Descemet membrane perforation is a well-described complication of DALK surgery. Studies report Descemet membrane perforation rates ranging from 4% to 39% [36,37,41]. However, micro-perforations of the Descemet membrane often do not prevent the completion of DALK. Nevertheless, there is a risk of postoperative Descemet membrane detachment and the formation of a pseudo-double anterior chamber [15]. In case of intraoperative Descemet membrane macro-perforation, unsuccessful or unsatisfactory lamellar dissection, or full-thickness opacity, another strategy to convert DALK to PK is to employ a mushroom-shaped graft [42,43].

Table 2 outlines the differential outcomes of studies on DALK for HSV keratitis.

### 3.2. Mushroom Keratoplasty (MK)

Two-piece mushroom keratoplasty has been proposed as a hybrid of PK and DALK. Postoperative outcomes for 2-piece MK compare favorably with the results of PK in eyes both at low- and high-risk for immunologic rejection [44].

To retain the advantages of DALK during conversion to PK, we have developed two-piece microkeratome-assisted mushroom keratoplasty (MK). The 2-piece MK graft consists of a large (9-mm-diameter) anterior lamella (mushroom ‘‘hat’’) and a smaller (6-mm-diameter) posterior lamella (mushroom ‘‘stem’’), which attaches to the anterior one without the need for sutures [45]. This design combines the advantages of minimal postoperative refractive errors typical of larger full-thickness grafts and the reduced risk of immunologic rejection related to the smaller antigenic load. In addition, the increased surface area of stromal contact between recipient and donor tissue that is obtained with the mushroom configuration theoretically speeds up wound healing compared to conventional PK [46].

In the setting of herpetic keratitis, two-piece MK can be performed when there is unsatisfactory clearance of the optical zone of a full-thickness opacity or a macroperforation of the Descemet membrane [13].

The standard surgical technique of two-piece MK is summarized as follows. After an initial 9.0-mm-diameter trephination and anterior lamellar dissection, the central 6.0-mm optical zone is excised at full thickness. Using a 250-μm microkeratome head, the donor cornea is split into anterior and posterior lamellae and punched to 9.0 mm and 6.0 mm, respectively. The donor posterior lamella is placed into the central hole of the recipient bed without sutures, while the donor anterior lamellar head is placed on top and sutured recipient bed using interrupted nylon 10–0 sutures [45].

In a study published in the *American Journal of Ophthalmology*, our group reported on 10-year outcomes of 52 eyes using the standard two-piece microkeratome-assisted mushroom keratoplasty and a predefined antiviral prophylaxis protocol. All eyes were defined as high risk for recurrence based on the presence of at least two quadrants of neovascularization. At 10 years, the graft junctions were imperceptible, with the cornea remodeling to its original form [43].

Another advantage of MK is its use of a large-diameter anterior lamellar graft that optimizes postoperative refractive outcomes [43]. MK for herpetic keratitis yielded a Snellen BSCVA of 20/40 at 1 year and 20/25 at 5 years, with 66% reaching 20/40 [43]. Our study compares favorably with reports on PK for herpetic keratitis showing a mean approximate Snellen BSCVA of 20/80 at 1 year and 20/50 at 5 years [47]. A higher mean endothelium cell density (ECD) was observed in our series (1155 cells/mm^2^) compared with values recorded in a series using PK performed for herpetic keratitis (approximately 1000 cells/mm^2^) at 5 years [48]. In terms of the ECL trend, the decline was greatest within the first year, as expected from surgically induced endothelial trauma, but within 4 years, ECD began to plateau with no significant changes between 4 and 10 years [43].

Moreover, the aim of MK is to limit the replacement of the recipient’s healthy endothelium, thereby reducing exposure to immunologic stimulation due to a lower density of antigen-presenting cells and a greater distance from limbal vessels. In addition to immunologic rejection and herpetic recurrence, long-term graft survival in herpetic corneal scars is also related to endothelial cell loss (ECL). MK reduces the transplantation to about 25% of the healthy recipient endothelium. This accounts for the early stabilization of ECL following two-piece MK, as the large reservoir of healthy endothelial cells in the recipient cornea can migrate across the surgical wound to the posterior surface of the graft if needed [43,44,45].

In the study by Yu, et al. [43], the 10-year cumulative risk of immune rejection was 9.7% and herpetic recurrence of 7.8%, with an overall graft survival of 92%. Additionally, in a previous study evaluating the outcomes of MK in corneas with postinfectious vascularized scars, 56% of which had post-herpetic scars, the use of this technique has resulted in a greater than 90% rate of 3-year graft survival [46]. In another recent study that described the long-term outcomes of keratoplasty in 120 eyes with herpetic scars, 18.3% of which required intraoperative conversion to 2-piece MK for macro-perforation of Descemet membrane or unsatisfactory stromal clearance of the optical zone, 9.0-mm DALK was successful in 81.7% of cases. Results showed that BSCVA did not differ significantly between converted MK and successful DALK at any time point. This result confirms that the optical outcome of the microkeratome-dissected interface in MK is comparable with that obtained by means of DALK by the big bubble technique. Furthermore, at 5 years, there were no significant differences in mean refractive astigmatism between the two groups (3.0 D in the MK group vs. 2.8 D in the DALK group). The postoperative complications in the two groups were as follows: the 5-year risk of immunologic rejection was 5% for MK and 3% for DALK (*p*  =  0.38); herpetic recurrence was 9% for MK and 6% for DALK (*p*  =  0.38); and graft failure was 5% for MK and 4% for DALK (*p*  =  0.75) [13].

The low rates of immunologic rejection, herpetic recurrence, and endothelial cell loss likely contributed to the excellent overall 5-year survival rates: 96% after DALK and 95% after MK, with no significant differences between the two procedures. These rates are favorable when compared to those reported by Wu et al. after conventional PK (78.8% at 5 years) and Halberstadt et al. (40.9% at 5 years), while they are in line with those reported for primary MK (96% at 5 years) [19,43,47].

Table 3 outlines the differential outcomes of studies on MK for HSV keratitis.

### 3.3. Postoperative Antiviral Prophylaxis

Antiviral prophylaxis has become the standard for preventing recurrence and consequently reducing failure of corneal grafts performed for herpetic scars [7,49]. This regimen prevents reactivation of latent HSV and reduces the risk of allograft rejection in vascularized corneas through downregulation of viral shedding and HSV-associated inflammation [7].

The Herpetic Eye Disease Study (HEDS) demonstrated the effectiveness of long-term oral acyclovir prophylaxis in preventing herpetic recurrence [6]. However, these trials excluded patients who had previously undergone corneal transplantation. Herpetic reactivation can be triggered immunologically by local trauma during surgery, which is a significant cause of corneal graft failure [50]. Therefore, the same acyclovir dosage used in the HEDS trials is likely not suitable for eyes undergoing keratoplasty for herpetic scars.

Various studies have shown the efficacy of long-term oral prophylaxis with acyclovir in reducing the rate of recurrent infections and graft failures following corneal transplantation for herpetic keratitis.

Nevertheless, there is still no consensus in terms of the optimal prophylactic regimen. A study by Simon et al. concluded that recurrences were more likely in patients on relatively lower doses (less than 800 mg daily) and in those who had undergone ocular surgery. Among patients experiencing recurrent HSV keratitis following ocular surgery, those on higher doses of oral acyclovir (an average dose of 1321 mg/day) experienced fewer recurrences compared to those on lower doses (average of 1000 mg/day [51].

In a study by Ghosh et al. comparing systemic and topical acyclovir for prevention of HSV keratitis recurrences after PK, there was a significant reduction in recurrence of HSV keratitis in patients treated with 400 mg oral acyclovir twice daily compared to those using 3% topical acyclovir at both one- and two-year follow-up [49].

A randomized clinical trial by Goldblum et al. also provided evidence supporting the efficacy of high-dose oral acyclovir (800 mg three to five times daily) with a gradual tapering over three years. However, no difference in rates of HSV recurrence, graft failure, or side effects between post-PK patients who received either valacyclovir (500 mg two to three times daily for four months and tapered to 250 mg twice daily for up to 30 months) or acyclovir (800 mg three to five doses per day and tapered to 400 mg twice daily for 11 to 36 months) were observed [52]. Based on the findings, HSV keratitis treatment guidelines by the American Academy of Ophthalmology strongly recommend using high-dose oral acyclovir (800 mg three times daily for at least one year) as antiviral prophylaxis after keratoplasty [7].

Our preferred regimen involves initial high-dose systemic and topical antiviral and steroid prophylaxis with extended taper, resulting in excellent visual outcomes and early stabilization of endothelial cell loss in eyes with vascularized herpetic corneal scars that underwent two-piece mushroom keratoplasty [43]. Although prolonged prophylaxis offers significant benefits, it poses the risk of selecting for HSV-1 strains that exhibit resistance to therapy. Of particular concern, long-term antiviral prophylaxis, as seen in several other chronic viral infections, is linked to the potential emergence of antiviral drug-resistant strains, notably acyclovir-resistant (ACVR) HSV-1 isolates [53]. The risk of inducing drug resistance with prolonged antiviral administration did not seem to affect surgical outcomes in this series [43]. Although no patient was tested specifically for the presence of antiviral-resistant HSV strains, based on our experience, the benefits of long-term prophylaxis to prevent the sight-threatening recurrence of herpetic keratitis significantly outweigh the risks.

Although our prophylactic regimen may be considered aggressive, prevention of rejection is of primary importance in achieving long-term graft survival. As aforementioned, the previously reported low rates of immune rejection and graft failure following DALK and two-piece MK with initial high-dose and extended taper of antiviral and steroid therapy underscores the benefits of prophylaxis to address the indefinite potential for both immunologic rejection and herpetic recurrence even years after corneal transplantation [13].

## 4. Conclusions

Recurrent HSV keratitis can be associated with permanent vision loss due to corneal scarring and astigmatism. Although many surgeons prefer not to perform keratoplasty in these eyes considered to be at high risk of graft failure, in recent years new surgical techniques have improved the outcomes of keratoplasty. DALK using the big-bubble technique is an excellent alternative to PK and should be used more commonly since it offers satisfactory visual outcome and stability for a longer period of time without the risk of endothelial rejection. Moreover, two-piece MK represents an excellent alternative to PK in case of unsuccessful or unsatisfactory lamellar dissection in attempted DALK surgery and has shown excellent visual results, relatively early stabilization of ECL, and reduced rates of herpetic recurrence, immunologic rejection, and graft failure. In any case, the use of postoperative antiviral therapy, especially with an initial high dose and prolonged tapering of systemic antivirals, is essential in preventing the recurrence of herpetic keratitis and the subsequent risk of graft failure.

## Figures and Tables

**Figure 1 viruses-16-01403-f001:**
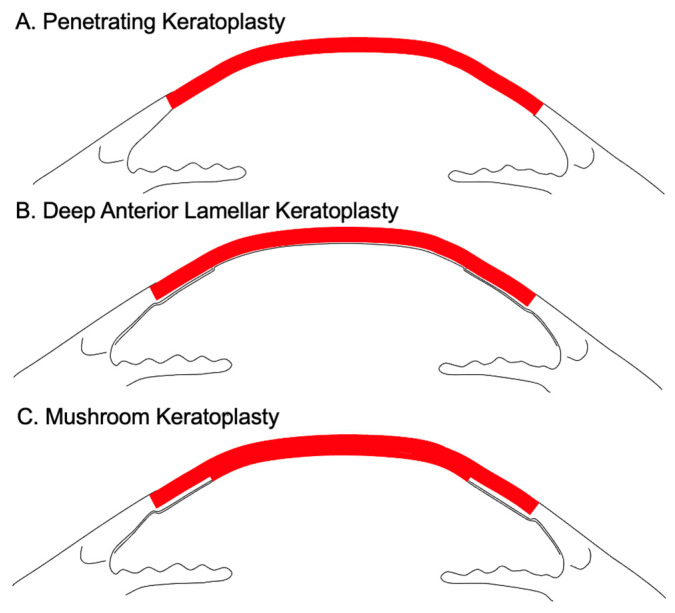
Schematic diagram of penetrating keratoplasty (**A**), deep anterior lamellar keratoplasty (**B**) and mushroom keratoplasty (**C**).

**Table 1 viruses-16-01403-t001:** Differential outcomes of studies on penetrating keratoplasty for HSV keratitis.

**Procedure Type**	PK	PK	PK	PK and DALK
**Author**	Lomholt et al. [17]	Goodfellow et al. [18]	Altay et al. [20]	Wu et al. [19]
**Year of publication**	1995	2011	2017	2012
**Total patients in the study**	72	403	55	121
**Treatment groups**	1 group	- PK without Postoperative Antiviral Prophylaxis: 152 eyes - PK + Topical Antiviral Prophylaxis: 65 eyes - PK + Oral Antiviral Prophylaxis: 186 eyes	- Group 1 (Quiescent herpetic corneal scar): 42 eyes - Group 2 (Corneal descemetocele or perforation): 13 eyes	- DALK: 58 eyes - PK: 63 eyes
**Pre/Postoperative Antiviral Prophylaxis**	No Prophylaxis	Topical or Oral Antiviral Prophylaxis (type and dosage not specified)	Oral Acyclovir 400 mg × 5/days for 3 months, then 400 mg × 2/days for 12 months	Oral Acyclovir 200 mg × 5/d for 3 months, then 400 mg 2 × d for a further 12 to 18 months
**Duration of follow-up**	24 months	60 months	Group 1: 20 months Group 2: 26 months	48 ± 27 months
**Outcomes**	Recurrence rate: 44% Rejection rate: 46%	Relative Risk of Graft Failure: - No medication: 1.0 - Topical only: 0.2 (CI 0.02–1.2, *p* = 0.07) - Oral medication: 0.3 (CI 0.2–0.7, *p* = 0.002)	- Group 1: Recurrence rate: 29% Rejection rate:10% Graft failure: 19% Visual Acuity: improvement in 93% eyes - Group 2: Recurrence rate: 31% Rejection rate: 23% Graft failure: 30% Visual Acuity: Improvement in 100% eyes	- PK group: Recurrence rate: 21% Rejection rate: 41% Graft failure: 22% Visual Acuity: Improvement in 51% eyes - DALK group: Recurrence rate: 10% Rejection rate: 0% Graft failure: 2% Visual Acuity: Improvement in 66% eyes
Legend: DALK = deep anterior lamellar keratoplasty; PK = penetrating keratoplasty				

**Table 2 viruses-16-01403-t002:** Differential outcomes of studies on deep anterior lamellar keratoplasty for HSV keratitis.

**Procedure Type**	DALK	DALK	DALK	DALK
**Author**	Sarnicola et al. [34]	Ren et al. [29]	Awan et al. [37]	Lyall et al. [38]
**Year of publication**	2010	2016	2014	2012
**Total patients in the study**	52	89	18	18
**Treatment groups**	1 group	3 groups Folds-off Group (*n* = 27) Folds-on Group (*n* = 14) No-folds Group (*n* = 48)	1 group	1 group
**Pre/Postoperative Antiviral Prophylaxis**	Oral Acyclovir 800 mg × 3/d for 2 months, tapering to 800 mg × 1/d for long term	Oral Acyclovir 400 mg × 5/d for 1 month, tapering to 400 mg × 2/d for 12 months	Oral Acyclovir × 2/d for 12 months	Oral Acyclovir 400 mg × 2/d for 12 months
**Duration of follow-up**	31 months	50 months	36 months	56 months
**Outcomes**	Recurrence rate: 0 % Rejection rate: 0 % Visual Acuity: - Preop. UVA 20/70 - Postop. UVA 20/40 Endothelial cell loss (6–12 months): 205 cells/mm^2^	Recurrence rate: 9% Rejection rate: 4.5% Visual Acuity: Mean preop. BCVA: logMAR 1.63 Postop.: - Folds-off Group: logMAR 0.42 - Folds-on Group: logMAR 0.48 - No-folds Group: logMAR 0.44	Recurrence rate (1 years): 5% Recurrence rate (3 years): 33% Rejection rate (3 years): 33% Graft failure (3 years): 17% Visual Acuity: logMAR 1.0 or better in 83%	Recurrence rate: 33% Rejection rate: 50% Graft failure: 28% Visual Acuity: - Preop. BCVA: logMAR 1.51 - Postop. BCVA: logMAR 0.82
**Procedure Type**	DALK	DALK	DALK	DALK
**Author**	Wang et al. [39]	Li, J. et al. [36]	Noble et al. [33]	Leccisotti et al. [35]
**Year of publication**	2012	2014	2007	2009
**Total patients in the study**	42 (43 eyes)	48	68	12
**Treatment groups**	Active Keratitis = 22 eyes Quiescent Keratitis = 21 eyes	2 groups DALK with Fresh corneal tissue (FTS): 27 eyes DALK with precut Anterior lamellar cap (ALC): 21 eyes	1 group (HSV Keratitis): 6 eyes 2 group (Other DALK Indications): 62 eyes	1 group
**Pre/Postoperative Antiviral Prophylaxis**	Preop.: i.v. Acyclovir 250 mg × 3/d for 3 d 0.15% Ganciclovir ointment × 1/d for 3 d 0.1% Acyclovir eye drops × 4/d for 3 d Postop.: 0.1% Acyclovir eye drops × 4/d for 6 months oral Acyclovir 400 mg × 3/d for 3 months	Oral Acyclovir 200 mg × 5/d for 3 months, tapering to 400 mg × 2/d for 12 months	No Prophylaxis	Oral acyclovir 400 mg × 3/d 1 month before surgery and continued for 5 months, then tapered to 800 mg × 1/d 6 months further.
**Duration of follow-up**	29 months	36 months	21 months	15 months
**Outcomes**	Recurrence rate: 14% - Active Keratitis Group: 18% - Quiescent Keratitis Group: 9.5% Rejection rate: 2.3% Graft failure: 2.3% Non-physiologic corneal graft endothelial cell loss or dysfunction was not observed. Postop. BSCVA - 20/100 to 20/40:86%	Recurrence rate: - FTS: 11% - ALC: 14% Rejection rate: 0% Visual Acuity: - Preop. BSCVA: logMAR 1.21 - Postop. BSCVA: - FTS: logMAR 0.26 - ALC: logMAR 0.28	HSV Keratitis Group: Recurrence rate: 33% Rejection rate: 0% Visual Acuity: Improvement in 84.9% of eyes	Recurrence rate: 0% Rejection rate: 0% Visual Acuity: BSCVA mean improvement 0.51
Legend: BCVA = best corrected visual acuity; DALK = deep anterior lamellar keratoplasty; HSV = herpes simplex virus; logMAR = logarithm of minimum angle of resolution; PK = penetrating keratoplasty; UVA =uncorrected visual acuity				

**Table 3 viruses-16-01403-t003:** Differential outcomes of studies on mushroom keratoplasty for HSV keratitis.

**Procedure Type**	MK	MK	MK and DALK
**Author**	Yu et al. [43]	Scorcia et al. [46]	Pellegrini et al. [13]
**Year of publication**	2020	2011	2023
**Total patients in the study**	52	31	122
**Treatment groups**	1 Group	- Herpetic Keratitis: 16 eyes - Bacterial Keratitis: 10 eyes - Acanthamebic Infection: 5 eyes	- DALK: 100 eyes - Converted MK: 22 eyes
**Pre/Postoperative Antiviral Prophylaxis**	Preop.: Month -3 to surgery: Oral Acyclovir 800 mg × 2/d; 0.15% Ganciclovir ointment × 1/d Postop.: Oral Acyclovir 800 mg × 5/d; 0.15% Ganciclovir ointment × 4/d for 2 weeks, then gradually tapered. Month 24 onwards: 0.15% Ganciclovir ointment × 1/d	HSV Keratitis Group: Oral Acyclovir 400 mg × 2/d for at least 1 year	Preop.: Month -3 to surgery: Oral Acyclovir 800 mg × 2/d; 0.15% Ganciclovir ointment × 1/d Postop.: Oral Acyclovir 800 mg × 5/d; 0.15% Ganciclovir ointment × 4/d for 2 weeks, then gradually tapered. Month 24 onwards: 0.15% Ganciclovir ointment × 1/d
**Duration of follow-up**	60–120 Months	36 months	54 ± 24 months
**Outcomes**	Recurrence rate: 7.8% Rejection rate: 9.7% Graft failure: 7.6% Visual Acuity: Preop BSCVA (1.71 ± 0.07) Postop BSCVA: 0.33 ± 0.33 (*p* < 0.001), 0.17 ± 0.18 (*p* = 0.016) at 1 and 2 years Endothelial Cell Loss rate: 41% ± 24%	HSV Keratitis Group: Recurrence rate: 0% Rejection rate: 0% All groups: Visual Acuity: 20/40 or better in 83.8% of eyes Endothelial Cell Loss rate: 23.5% at 12 months, 34.5% at 24 months, and 40.7% at 36 months.	Recurrence rate: - DALK: 6% - MK: 9% Rejection rate: - DALK: 3% - MK: 5% Graft failure: - DALK: 4% - MK: 5% Visual Acuity (BSCVA (logMAR)): - DALK: 0.10 ± 0.12 - MK: 0.09 ± 0.15 (*p* = 0.75) Mean Endothelial Cell Loss rate: - DALK: 4.2% - MK: 10.9%
Legend: BSCVA = best spectacle corrected visual acuity; DALK = deep anterior lamellar keratoplasty; HSV = herpes simplex virus; PK = penetrating keratoplasty			

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
