# Peer review of "Outcomes of Corneal Transplantation for Herpetic Keratitis: A Narrative Review"

_viruses, 2024, doi:10.3390/v16091403_

Round 1

Reviewer 1 Report

Comments and Suggestions for Authors

The MS submitted by Nardella M. and colleagues reviews the literature on the outcome of different cornea transplant techniques performed on patients with herpetic keratitis to restore corneal function. This review is well-written and nicely compares the differences in outcome described for the 3 main cornea transplant techniques: PK, DALK & MK. Finally, benefit of antiviral prophylaxis to lower the risk of corneal HSV reactivation is well-described.

However, several changes are recommended to improve the manuscript:

1.      Introduction section (1/1): Provide background of HSV-1 latency focused on the cornea and how this is controlled. Furthermore, n=555 articles found, which time span included, and why did the authors remove about 90% of these studies for their review?  

2.      Discussion section (1/5): the readership of this journal may not be aware of the cornea transplant technologies described. Add a figure to clarify these techniques schematically: what is excised/transplanted?

3.      Discussion section (2/5): The ‘summing up’ of differential outcomes between studies and techniques is of course necessary, but hard to read and interpret by the reader. A Table, but ideally a multi-panel Figure is preferred to present the main differences in specific outcomes between the studies reviewed. Figure panels for each cornea transplant technique is preferred.

4.      Discussion section (3/5): The table included at the end of the Discussion section shows the antiviral prophylactic regimen the authors developed. Unclear if this has already been published, reference 42, if so duplication own work is not preferred.

5.      Discussion section (4/5): The table included at the end of the Discussion section shows the antiviral prophylactic regimen the authors developed. Unclear if this has already been published, reference 42, if so duplication own work is not preferred.

6.      Discussion section (5/5): antiviral prophylaxis, especially for >0.5 yr, comes with a price: therapy refractory HSV-1 strains. This needs to be discussed in more detail and how to overcome this major drawback of prophylaxis.

Comments on the Quality of English Language

NA

Author Response

  1.    Reviewer 1:  Introduction section (1/1): Provide background of HSV-1 latency focused on the cornea and how this is controlled.

Response:

The following statements have been added to the introduction (Page 1, Lines 43-51). HSV type-1 establishes latency in the trigeminal ganglia following primary infection of the cornea. HSV-1 virus remains dormant in neuronal cells, avoiding immune detection. Reactivation of HSV-1 due to various triggers such as stress or immunosuppression can lead to recurrent corneal infections.7 Control of HSV-1 latency and reactivation is largely mediated by the host's immune system, particularly through the actions of T-cells (particularly CD8+ T cells) and cytokines that suppress viral gene expression and maintain the virus in a quiescent state.8

  1. Furthermore, n=555 articles found, which time span included, and why did the authors remove about 90% of these studies for their review?  
    Response:

We have edited the Materials and Methods section to specify the time span and to clarify that the articles were screened based on the aforementioned inclusion/ exclusion criteria  (Page 2, Lines 69-79)

  1.      Discussion section (1/5): the readership of this journal may not be aware of the cornea transplant technologies described. Add a figure to clarify these techniques schematically: what is excised/transplanted?
    Response:

We have added the schematic diagram of the different techniques as Figure 1 and the following text in discussion to clarify the techniques  (Page 2, Lines 81-87). Penetrating keratoplasty (PK) involves the full-thickness transplantation of a donor cornea to replace the damaged corneal tissue (Figure 1A).14 On the other hand, deep anterior lamellar keratoplasty involves selective replacement of diseased corneal stroma with the anterior donor lamella (Figure 1B). Mushroom keratoplasty is a form of penetrating keratoplasty which involves minimal endothelial replacement with transplantation of a large 9mm diameter anterior lamella and a smaller 6mm diameter posterior lamella  (Figure 1C).

  1.      Discussion section (2/5): The ‘summing up’ of differential outcomes between studies and techniques is of course necessary, but hard to read and interpret by the reader. A Table, but ideally a multi-panel Figure is preferred to present the main differences in specific outcomes between the studies reviewed. Figure panels for each cornea transplant technique is preferred.
    Response:

We have added table 1 (Pages 4-5), table 2 (Pages 7-9) and table 3 (Pages 11-12) to present differential outcomes for each technique, penetrating keratoplasty, deep anterior lamellar keratoplasty and mushroom keratoplasty, respectively

  1.      Discussion section (3/5): The table included at the end of the Discussion section shows the antiviral prophylactic regimen the authors developed. Unclear if this has already been published, reference 42, if so duplication own work is not preferred.
    Response:

We have removed this table

  1.      Discussion section (4/5): The table included at the end of the Discussion section shows the antiviral prophylactic regimen the authors developed. Unclear if this has already been published, reference 42, if so duplication own work is not preferred.
    Response:

We have removed this table

  1.      Discussion section (5/5): antiviral prophylaxis, especially for >0.5 yr, comes with a price: therapy refractory HSV-1 strains. This needs to be discussed in more detail and how to overcome this major drawback of prophylaxis.
    Response:

We have added the following statement in the discussion  (Page 13, Lines 367-379)

Our preferred regimen involves initial high dose systemic and topical antiviral and steroid prophylaxis with extended taper resulting in excellent visual outcomes and early stabilization of endothelial cell loss in eyes with vascularized herpetic corneal scars that underwent two-piece mushroom keratoplasty.43 Although prolonged prophylaxis offers significant benefits, it poses the risk of selecting for HSV-1 strains that exhibit resistance to therapy. Of particular concern, long-term antiviral prophylaxis, as seen in several other chronic viral infections, is linked to the potential emergence of antiviral drug-resistant strains, notably acyclovir-resistant (ACVR) HSV-1 isolates.53 The risk of inducing drug resistance with prolonged antiviral administration did not seem to affect surgical outcomes in this series. 43 Although no patient was tested specifically for the presence of antiviral-resistant HSV strains, based on our experience, the benefits of long-term prophylaxis to prevent sight-threatening recurrence of herpetic keratitis significantly outweigh the risks.

Reviewer 2 Report

Comments and Suggestions for Authors

This paper is a clear review of the topic, very helpful for those not familiar with it. It only requires minor changes, most of which related with text inaccuracies.

Lines 30-31 This sentence is unclear

Check the text for several minor inaccuracies. Examples in Line 86, 97, 12, 150, 159-169, 174,176, 322-325, 328-330  

Line 260: yelded ! 263 reference needed

263-265  The authors did not state the paper was one of theirs

265 reference needed

274-275 add reference about cell migration

278-300 All this section should be re-written not as a DALK/MK comparison, but as a MK/DALK comparison.

279 add: in the study by Yu et al. 42.

282-284 Re-phrase talking about the mushroom converted eyes first. The current sentence is confusing.

286 change “successful DALK and converted MK” into “converted MK and successful DALK”

Comments on the Quality of English Language

Minor editing required.

Author Response

Reviewer 2: This paper is a clear review of the topic, very helpful for those not familiar with it. It only requires minor changes, most of which related with text inaccuracies.
Lines 30-31 This sentence is unclear
Response: We have revised the statement as follows (Page 1, Lines 30-31): Majority of primary HSV infections are asymptomatic or present with nonspecific oropharyngeal symptoms 3,4.

  1. Check the text for several minor inaccuracies. Examples in Line 86, 97, 12, 150, 159-169, 174,176, 322-325, 328-330  
    Response: 

Line 86 in the original manuscript now reads as follows (Page 3, Lines 97-99): In study by Lomholt et al, the recurrence rate of HSV keratitis in patients who received PK with no antiviral prophylaxis was 44% within the first 2 years postoperatively. 17

Line 97 in the original manuscript now reads as follows (Page 3, Lines 109-111): Moreover, endothelial rejection was remarkably more common than stromal rejection.

Line 12 in the original manuscript now reads as follows (Page 1, Lines 11-12): Although most cases can be successfully managed with medical therapy, HSV keratitis associated with visually significant stromal scarring often requires corneal transplantation for visual rehabilitation

Line 150 in the original manuscript now reads as follows: (Page 6, Lines 163-164) On the other hand, studies have shown that stromal rejection is a rare complication of DALK surgery

Line 159-169 in the original manuscript now reads as follows(Page 6, Lines 172-182): A smaller series including 18 eyes by Awan et al. demonstrated a graft rejection rate of 33% at three years, which still compares favorably to the reported rates following PK. 37 In the same series, the recurrence of HSV keratitis was 5% in first year, which is significantly less in comparison to those reported for PK. 37 In addition, 3-year graft survival rate was 83%, which is higher than the 2-year survival rate of 67% as reported by Lomholt et al.17 Slightly worse results were obtained by Lyall et al. with a graft survival rate of 72% and a higher HSV recurrence rate (33%), probably due to a use of a lower dose of oral acyclovir prophylaxis (400 mg twice daily).38 These observations suggest oral acyclovir may minimize the risk but does not completely eliminate the recurrence of HSV keratitis after DALK. Furthermore, low dose of acyclovir after surgery with the non-use of topical steroids and antibiotics has also been associated with graft failure or graft melting.36

Line 174 in the original manuscript now reads as follows (Page 7, Lines 216-219): Additionally, the 9.0-mm DALK group experienced significantly less astigmatism, with an average postoperative refractive astigmatism that was 1 D lower than those observed in the 8.0-mm group
Line 176 now reads as follows (Page 6, Lines 183-187): A study by Wang et al. evaluated the outcomes of DALK in patients with herpetic stromal keratitis.39

Line 322-325 in the original manuscript now reads as follows  (Page 12, Lines 353-356):: In a study by Ghosh et al. comparing systemic and topical acyclovir for prevention of HSV keratitis recurrences after PK, there was a significant reduction in recurrence of HSV keratitis in patients treated with 400 mg oral acyclovir twice daily compared to those using 3% topical acyclovir at both one- and two- year follow-up. 49

Line 328-330 in the original manuscript now reads as follows  (Page 12, Lines 359-363):: However, no difference in rates of HSV recurrence, graft failure, or side effects between post-PK patients who received either valacyclovir (500 mg two to three times daily for four months and tapered to 250 mg twice daily for up to 30 months) or acyclovir (800 mg three to five doses per day and tapered to 400 mg twice daily for 11 to 36 months) were observed..52

  1. Line 260: yelded !
    Response:

 Line 260 in the original manuscript (Page 10, Line 287) has been revised as “yielded

  1. 263 reference needed

Response:

Reference has been added. 48. 48. Halberstadt M. The outcome of corneal grafting in patients with stromal keratitis of herpetic and non-herpetic origin. Br. J. Ophthalmol. 86, 646–652 (2002).

  1. 263-265  The authors did not state the paper was one of theirs

 Response:

We have revised and mention that this was “our study” (Page 10, Line 288)

  1.  

    265 reference needed
    Response: 

Reference has been added. 52. Goldblum D, Bachmann C, Tappeiner C, Garweg J, Frueh BE. Comparison of oral antiviral therapy with valacyclovir or acyclovir after penetrating keratoplasty for herpetic keratitis. Br. J. Ophthalmol. 92, 1201–1205 (2008).

14. 274-275 add reference about cell migration
Response: Reference about cell migration has been added : Reinhard, T., et al., [Chronic endothelial cell loss of the graft after penetrating keratoplasty: influence of endothelial cell migration from graft to host]. Klin Monbl Augenheilkd, 2002. 219(6): p. 410-6.

15. 278-300 All this section should be re-written not as a DALK/MK comparison, but as a MK/DALK comparison.
Response: 

We have re-written Lines 278-300 in the original manuscript (Page 10, Lines 305-320) as an MK/DALK comparison.

16. 279 add: in the study by Yu et al. 42.
Response: 

We have added the phrase “In the study by Yu et al.,42” (Page 10, Line 305)

17. 282-284 Re-phrase talking about the mushroom converted eyes first. The current sentence is confusing.
Response:

We have revised the statement which now reads as “In another recent study that described the long-term outcomes of keratoplasty in 120 eyes with herpetic scars, 18.3% of which required intraoperative conversion to 2-piece MK for macro-perforation of Descemet membrane or unsatisfactory stromal clearance of the optical zone while 9.0-mm DALK was successful in 81.7% of cases.” (Page 10, Lines 309-313)

  1. 286 change “successful DALK and converted MK” into “converted MK and successful DALK”
    Response:

We have revised the statement as “converted MK and successful DALK” (Page 10, 314)